# Performance Evaluation of Nested Polymerase Chain Reaction (Nested PCR), Light Microscopy, and *Plasmodium falciparum* Histidine-Rich Protein 2 Rapid Diagnostic Test (PfHRP2 RDT) in the Detection of Falciparum Malaria in a High-Transmission Setting in Southwestern Nigeria

**DOI:** 10.3390/pathogens11111312

**Published:** 2022-11-09

**Authors:** Oluwaseun Bunmi Awosolu, Zary Shariman Yahaya, Meor Termizi Farah Haziqah, Titus Adeniyi Olusi

**Affiliations:** 1School of Biological Sciences, Universiti Sains Malaysia, Penang 11800, Malaysia; 2Vector Control Research Unit, Universiti Sains Malaysia, Minden, Penang 11800, Malaysia; 3Department of Biology, Federal University of Technology, Akure 340252, Nigeria

**Keywords:** accuracy, Cohen’s kappa, predictive value, sensitivity, specificity

## Abstract

Malaria remains a major public health challenge worldwide. In order to ensure a prompt and accurate malaria diagnosis, the World Health Organization recommended the confirmatory parasitological diagnosis of malaria by microscopy and malaria rapid diagnostic test (RDT) prior to antimalarial administration and treatment. This study was designed to evaluate the performance of nested polymerase chain reaction (nested PCR), light microscopy, and *Plasmodium falciparum* histidine-rich protein 2 rapid diagnostic test (PfHRP2 RDT) in the detection of falciparum malaria in Akure, Nigeria. A cross-sectional and hospital-based study involving 601 febrile volunteer participants was conducted in Akure, Nigeria. Approximately 2–3 mL venous blood samples were obtained from each study participant for parasitological confirmation by microscopy and PfHRP2-based malaria RDT. Thick and thin films were prepared and viewed under the light microscope for parasite detection, parasite density quantification, and species identification, respectively. Dry blood spot samples were prepared on 3MM Whatman filter paper for nested PCR. The overall prevalence of microscopy, PfHRP2 RDT, and nested PCR were 64.89% (390/601), 65.7% (395/601), and 67.39% (405/601), respectively. The estimates of sensitivity, specificity, positive predictive value, negative predictive value, accuracy, and Youden’s j index of microscopy and RDT were 96.30, 100.00, 100.00, 92.89, 97.50, 0.963, and 95.06, 94.90, 97.47, 90.29, 95.01, and 0.899, respectively. Malaria RDT recorded higher false negativity, compared microscopy (4.94% vs. 3.70%). A near perfect agreement was reported between microscopy and nested PCR, and between PfHRP2 RDT and nested PCR with Cohen’s kappa (k) values of 0.94 and 0.88, respectively. This study revealed that PfHRP2 RDT and microscopy continues to remain sensitive and specific for falciparum malaria diagnosis in the study area.

## 1. Introduction

Malaria is one of the deadliest infectious diseases worldwide, particularly in tropical and sub-tropical regions of the world, including Nigeria. Meanwhile, the prompt treatment control, management, and elimination of malaria disease is basically predicated on early and accurate detection of malaria parasites in infected individuals [1,2]. However, malaria diagnosis remains a significant problem because the performance of diagnostic tools varies in different epidemiological settings [3]. In order to ensure proper, accurate, and rapid malaria diagnosis for prompt treatment and management control, the World Health Organization (WHO) recommended the confirmatory parasitological diagnosis of malaria by microscopy, which remains the gold standard, and malaria RDT for point of care diagnosis prior antimalarial administration and treatment [4]. In view of this, both microscopy and malaria histidine-rich protein-2 RDT have been recommended and approved for malaria diagnosis by the Nigerian malarial control policy into the Nigeria national malaria treatment guideline for the diagnosis of malaria disease in Nigeria.

Apparently, microscopy is advantageous in detecting and quantifying malaria parasite species, and it is quite cheap and excellent when used by an adequately trained microscopist [5]. However, microscopy may be inefficient in detecting submicroscopic malaria infection, though depending on the competency level of the microscopist [4,6]. Similarly, microscopy is subjective to the microscopist’s ability when used for the identification and counting of parasite density (6). Moreover, microscopy could be burdensome, as it requires a lot of work and energy, coupled with the fact that it is time-consuming and requires high expertise and training. Thus, generally, in most African countries, the capacity for good malaria microscopy is relatively low, and this is a major challenge in effective malaria case management. Unlike microscopy, malaria RDT is fast and easy to use. Popular malaria antigens employed in malaria RDTs are histidine-rich protein-2 (HRP2), *Plasmodium* lactate dehydrogenase (pLDH), and the aldolase [7]. One major difference is that histidine-rich protein-2 is an antigen that is specific for the detection of *P. falciparum*. However, *Plasmodium* lactate dehydrogenase (pLDH) and aldolase can be employed to detect all the four species of *Plasmodium* that infect humans, though they are inadequate for differentiating between *P. vivax, P. malariae*, and *P. ovale*, hence the name pan-pLDH and pan-aldolase [7,8]. Rapid diagnostic tests (RDTs) that detect pLDH do not produce persistently positive results after malaria treatment, as in the case of malaria histidine-rich protein 2 test. Although malaria RDT is rapid and easy to employ in malaria detection, it has been observed to be inefficacious in detecting malaria disease with low parasitaemia levels, with a detection limit of ≥100 parasite/μL of blood [9,10]. Additionally, false-positive results have been documented as a result of malaria antigen persistence in the blood, even after successful treatment. Thus, PfHRP2 RDT is practically unable to differentiate between past and current malaria infection. Similarly, false negative has been noted due to *P. falciparum* histidine-rich protein 2 (*pfhrp*2) gene deletions [11]. Therefore, there is a high probability of underdiagnosis, overdiagnosis, and misdiagnosis, which can eventually lead to undertreatment, overtreatment, and mistreatment [12].

Regarding nested PCR, it is most suitable for field research, and this technique has been confirmed by several studies to be highly sensitive and specific, such that it can detect a submicroscopic malaria parasite with a low density of 10 parasites/μL of blood [13]. The more sensitive methods of loop-mediated isothermal amplification (LAMP) and nucleic acid sequence-based amplification (NASBA) can achieve detection limits of 5–10 parasite/μL and <1 parasite/μL of blood, respectively [13]. Though nested PCR is highly sensitive and specific, it requires expensive reagents and a heavy laboratory setup with high financial implications that remain unattainable in resource-poor, malaria-endemic settings. However, due to its high sensitivity and specificity, it can be employed as a standard for comparison while testing and monitoring the efficacy of malaria RDT and microscopy.

Obviously, based on plethora of reports from previous studies conducted in malaria-endemic settings, it is well-established that malaria RDT and, to an extent, microscopy have low performance for the detection of submicroscopic malaria infections [7,14]. For instance, the sensitivities of 87.5% for malaria RDT and 37.5% for microscopy were reported in Lagos and Edo states, Nigeria [12]. Similarly, a sensitivity of 29% and a specificity of 89% were reported in a hospital in Asaba, Southern part of Nigeria [15]. Thus, in order to ensure proper malaria control and elimination, there is a need for constant evaluation of the performance of malaria RDT and microscopy, in comparison with a more sensitive diagnostic tool, such as polymerase chain reaction. This will assist in determining the accuracy of the recommended diagnostic tools and inform proper and appropriate policy regarding malaria diagnosis in Nigeria [16]. Thus, this study aimed to evaluate the performance efficacy of polymerase chain reaction, microscopy, and malaria histidine-rich protein 2 RDT in the detection of falciparum malaria in Akure, Nigeria.

## 2. Materials and Methods

### 2.1. Study Area

The study was carried out in some selected health facilities in Akure South local government area (LGA) of Ondo State, Southwestern Nigeria. The selected health facilities include Orita Obele Basic Health Centre, FUTA University Health Centre, State Specialist General Hospital, and Don Bosco Medical Centre. Akure South LGA is the most populous LGA in Ondo State, with an estimated population projection of 471,100 people in 2015 [17]. It is located within Akure city, the capital of Ondo State (Figure 1). Akure South LGA is located between latitude 7°12′18″ N and longitude 5°11′16″ E. Generally, the weather is typical of tropical regions. The local climate is made up of the dry season and rainy season, spanning from November to March and April to October, respectively. The average annual rainfall is 2378 mm, with temperatures ranging from 25.2 °C to 28.1 °C and relative humidity of about 80% [18]. Malaria is hyperendemic in Akure, with *P. falciparum* as the dominant species [19].

### 2.2. Study Design, Study Population and Sample Size Determination

This research was a cross-sectional and hospital-based study conducted between October 2019 and February 2020. There were four randomly selected health facilities, consisting of Orita Obele Basic Health Centre, FUTA University Health Centre, State Specialist General Hospital, and Don Bosco Medical Centre. A face-to-face interview was employed to collect important information, such as age, sex, and occupation, from each of the volunteered individuals visiting the health facilities. Recruitment of participants was carried out in the outpatient section of the health facilities, with the assistance of health personnel. The inclusion criteria were fever with temperature ≥37.5 °C, resident within the community for a minimum of 6 months, completing the questionnaire, submission of the blood sample, and readiness to provide informed consent. Generally, all study participants recruited were individuals presenting with symptoms suggestive of malaria infection, as determined by the assisting health officer. Out of the 615 potential participants initially selected, only 601 individuals eventually satisfied the inclusion criteria and participated in the study, thereby giving a response rate of 97.72%. The participants’ recruitment flow chart is shown in Figure 2.

Regarding the study sample size, it was computed by using the single population proportion estimate formula of Araoye: N = t^2^ × P (1 − P)/M^2^, where N is the sample size, t is the normal standard deviation, which is 1.96, P is the previous malaria prevalence around the study area, and M is the degree of precision, which is 5% (or 0.05) [20]. By employing a prior report of malaria prevalence of 71.7% (0.717) in Akure [19] and a confidence interval (C.I) of 95% with a precision level of 5%, a minimum of 312 participants were needed for the study. Notwithstanding, 601 participants were randomly selected in the study to accommodate any statistical error.

### 2.3. Collection of Blood Samples

The blood samples were collected intravenously with the assistance of a well-trained laboratory technician. Approximately 2–3 mL venous blood was collected from each volunteer participant. Thereafter, the blood samples were transferred into ethylenediaminetetraacetic acid (EDTA) tube to prevent the coagulation of the blood samples. Additionally, 2–3 drops of blood were aliquoted on 3 MM Whatman^®^ filter paper (Whatman International Ltd., Maidstone, UK), air-dried in a dust-free area, wrapped in plastic sample bags, placed in sealed zip-lock plastic bags with silica gel to prevent DNA degradation, and stored at room temperature until use.

### 2.4. Microscopy Examination

Thick and thin blood films were prepared from the blood in EDTA for the detection and quantification of malaria parasite species [21]. The thin smear was fixed in absolute ethanol. Subsequently, a 10% Giemsa stain (GIBCO, Scotland, UK) at pH 7.2. was added to both the thick and thin smears on the slide for 15 min and was washed with distilled water. The slides were subsequently air-dried in the laboratory. The light microscope was employed to view the slides using the ×100 oil immersion objective lens to determine the *Plasmodium* parasites prevalence, parasite density, and their exact species. The thin film was used in the identification of *Plasmodium* species, while parasite density was determined using the thick smear. Each slide was assumed to be negative when no parasite was observed after approximately 200 microscopic oil immersion fields have been observed. Additionally, all the slides were stored in secured slide-boxes, and quality assurance was carried out on 10% of all the positive slides by an expert laboratory microscopist for species identification and confirmation, and there was no difference after the quality assurance was performed. Afterwards, all the blood samples were confirmed by nested PCR. Moreover, all participants positive for falciparum malaria were treated according to national standards and guidelines with artemether-lumefantrine (AL).

### 2.5. Parasite Density Estimation

Parasite density was calculated as the number of parasite/µL of blood, assuming an average leucocyte count of 8000/µL of blood for an average individual [1]. Therefore, the parasite density was calculated from a thick blood smear, according to WHO guidelines, by calculating the number of asexual parasites × 8000/number of WBC counted. The parasite density was categorized into three: low (parasitaemia <1000 parasites/μL of blood), moderate (parasitaemia between 1000 to 9999 parasites/μL of blood), and severe (parasitaemia ≥10,000 parasites/μL of blood) [22].

### 2.6. Malaria Rapid Diagnostic Tests (RDTs)

The CareStart™ PfHRP2 RDT (Access BIO, Inc., Monmouth Junction, NJ, USA; year of manufacture: 2018) was employed for the rapid detection of the *P. falciparum* histidine-rich protein 2 antigen in blood samples by following the manufacturer’s instructions. The test kits were kept at an optimal temperature between 2–30 °C, and the results were read after 20–25 min. Briefly, approximately 5 µL from each blood sample was aliquoted on the sample window of the PfHRP2 RDT cassette. Thereafter, three drops of diluent were added. The PfHRP2 RDT was left intact at room temperature for a period of 20 min, and the results were recorded as positive or negative as appropriate after checking the cassette for the presence of distinct lines.

### 2.7. DNA Extraction and Molecular Analysis

Genomic DNA was extracted from the 3 MM Whatman filter paper dried blood spot (DBS) using the DNeasy^®^ Blood and Tissue Kit, cat. No. 69506 (QIAGEN, Hilden, Germany) by following the manufacturer’s instructions. Moreover, genomic DNA was eluted in 50 mL and was kept at −20 °C until nested PCR analysis. Regarding the nested PCR, *Plasmodium* genus-specific and species-specific primers targeting the 18 small subunits ribosomal RNA (18S rRNA) genes were designed according to Snounou and Singh [23], with minor modifications in the cycling conditions. The nested PCR amplification was carried out in a total reaction volume of 50 μL, which included 1 μL each of the genus-specific primers of rPLU1 (forward primer) and rPLU5 (reverse primer), 2 μL of extracted gDNA template, 5 μL of nested PCR buffer, 4 μL of dNTPs, 36 μL of double-distilled water (ddH2O), and 1 U of Taq DNA polymerase. The amplicon of the primary reaction was used in the nested PCR reaction as a template. The 3D7 strains of *Plasmodium* culture were used as the positive control, while ultra-pure water was employed as the negative control. The primers used are shown in Table 1.

The electrophoresis of the secondary nested PCR amplification products was carried out in a 2% agarose and 0.5X Tris-acetate EDTA (0.5X TAE buffer). Thereafter, the gel was stained with ethidium bromide and subsequently visualized using the ultraviolet–violet (UV) gel documentation system (GelDoc, Biorad, Hercules, CA, USA). The size of the *Plasmodium* nested PCR product was measured using a 50 base pairs ladder. The nested PCR products of 205 base pairs, as observed on the gel, were thereafter purified and sequenced.

### 2.8. Statistical Analyses

Data entry and analyses were conducted through Statistical Package for Social Sciences (SPSS), version 22.0 (IBM corporation, NY, USA). The presence or absence of malaria disease was computed, and the variance in the prevalence between the age groups and sex was calculated using a chi-square test at 95% confidence level. Statistical significance was set at *p*-values ≤ 0.05. The sensitivity, specificity, positive predictive value (PPV), negative predictive value (NPV), diagnostic accuracy, and Youden index (J) were calculated by using the GraphPad Prism (version 8.4.3 for Mac, GraphPad Software, San Diego, California, USA). Computations were carried out using the contingency table. The total accuracy was computed through the formula: Accuracy = (TP + TN)/(TP + FP + FN + TN), where TP = True Positive, TN = True Negative, FN = False Negative, and FP = False-Positive.

### 2.9. Ethical Considerations 

Prior to the commencement of the study, the ethical protocol was reviewed, and approval was obtained from the Ondo State Ministry of Health, with reference number OSHREC/10/01/2019/082, and the Ethical Review Committee of the Federal University of Technology, Akure, Nigeria. Additionally, permission was sought from the hospital management board of all the health facilities in the study areas. Written informed consent and accent were sought from each of the participants and caregivers or guardians after a detailed explanation of the study protocol and procedures, coupled with the risk and benefits of the study. All confirmed malaria-positive individuals were treated according to the national guidelines. This study was conducted in accordance with the Helsinki Declaration of 1975 and as amended in the year 2000.

## 3. Results

### 3.1. Distribution of the Study Participants

Out of the 615 potential participants enrolled on the study, only 601 (97.72%) met the criteria for inclusion in the study. Therefore, a total of 601 individuals were screened for malaria parasite using PfHRP2 RDT, microscopy, and nested PCR, of which males were 341 (56.7%), while females were 260 (43.3%). Additionally, with respect to the age group, the highest age group examined was ≥20 years, with 310 (51.6%), followed by the age group 13–19 years, with 230 (38.3%), while the least age group examined in this study was ≤12 years, with 61 (10.1%) participants. Similarly, the study sites and the number of participants included Orita Obele Health Centre, 100 (16.64%); FUTA Health Centre, 203 (33.78); State Specialist Hospital, 197 (32.78%); and Don Bosco Health Centre, 101 (16.81%) (Table 2).

### 3.2. Malaria Prevalence by Microscopy, pfhrp 2 RDT and Nested PCR

The malaria prevalence by microscopy, PfHRP2 RDT, and nested PCR were 390 (64.89%), 395 (65.72%), and 405 (67.38%), respectively. A total of 385 samples were observed to be positive by all three diagnostic techniques. However, 15 samples were positive only by nested PCR and 10 samples only by PfHRP2 RDT. No sample was detected by microscopy only. The Venn diagram comparing the proportion of prevalence of *P. falciparum* by microscopy, PfHRP2 RDT, and nested PCR techniques is shown in Figure 3.

### 3.3. Diagnostic Performance of pfhrp 2 RDT and Microscopy Using Nested PCR as Reference Standard

Table 3 details the malaria prevalence by microscopy and PfHRP2 RDT, with respect to nested PCR as a reference standard. The TP and TN for microscopy were 390 and 196, while the TP and TN for PfHRP2 RDT were 385 and 186, respectively. 

Moreover, Table 4 presents the diagnostic performance of microscopy and PfHRP2 RDT by using nested PCR as a reference standard. When nested PCR was considered as the reference standard, the estimates of sensitivity, specificity, positive predictive value (PPV), negative predictive value (NPV), accuracy, Youden’s j index, Cohen’s kappa (K) of microscopy, and PfHRP2 RDT were 96.30, 100.00, 100.00, 92.89, 97.50, 0.963, 0.94 and 95.06, 94.90, 97.47, 90.29, 95.01, 0.8996, and 0.88, respectively. It is obvious that the sensitivity, specificity, and accuracy of microscopy were greater than that of PfHRP2 RDT. The degree of agreement between microscopy, PfHRP2 RDT, and nested PCR, as measured by Cohen’s kappa (K), was near perfect (0.94 vs. 0.88). Similarly, the false positivity rates of microscopy and PfHRP2 RDT were 0 and 5.10%, while the false negativity rates of microscopy and PfHRP2 RDT were 3.70% and 4.94%. 

Table 5 details the sensitivity, specificity, and predictive values of microscopy and PfHRP2 RDT by age group with nested PCR as the reference standard. The sensitivity and specificity of microscopy significantly (*p* < 0.0001) decreased as age increased. The age group ≤12 years had the highest sensitivity and specificity (100.00 (92.44–100.00) vs. 100.00 (78.47–100.00)). However, the age group ≥20 years had the least sensitivity of 94.39% (C.I: 90.23–96.84) and a high specificity of 100% (C.I: 96.74–100.00). Similarly, the sensitivity and specificity of PfHRP2 RDT significantly (*p* < 0.0001) decreased with an increase in age. The age group ≤12 years significantly (*p* < 0.0001) had the highest sensitivity and specificity compared to the age group ≥20 years, which had the least sensitivity and specificity.

Table 6 reveals the malaria prevalence by microscopy, PfHRP2 RDT, and nested PCR, in relation to the parasite density classification. Obviously, nested PCR alone detected 15 submicroscopic malaria infections that were not detected by both microscopy and PfHRP2 RDT. Similarly, both microscopy and nested PCR detected five samples with low parasite densities of 51–100 parasites/μL that were not detected by PfHRP2 RDT.

## 4. Discussion

The findings from this study highlight the high performance of microscopy and histidine-rich protein 2 RDT, with respect to nested PCR, which was employed as the standard reference. The World Health Organization (WHO) recommended that all suspected cases of malaria should be promptly and accurately confirmed by RDT and microscopy, which is considered the gold standard technique [4,24]. Similarly, the Nigeria Federal Ministry of Health (FMOH) recommended and approved the use of both microscopy and malaria RDT in health facilities for malaria diagnosis in Nigeria [25]. Therefore, currently in Nigeria, the major diagnostic tools employed in detecting malaria infection among malaria-infected patients include microscopy and histidine-rich protein 2-based malaria RDT. However, proper malaria control and elimination require a more sensitive detection tool, which is the nested PCR technique [26,27]. 

Apparently, this study revealed that nested PCR significantly detected a higher malaria prevalence (67.39%), compared to microscopy (64.89%) and RDT (65.72%). Studies have shown that nested PCR can detect a low parasite density of 10 parasites/μL of blood [10,28]. The observation that nested PCR has the highest level of malaria parasite detection in our study is very similar to the reports of studies conducted in other parts of Nigeria [12,29,30] and other high and low malaria endemic settings across the world, such as Ghana [31,32], Ethiopia [33,34], Democratic Republic of Congo [35], and India [36,37]. Furthermore, in agreement with our findings, Kiyonga Aimeé [38] unequivocally demonstrated that nested PCR detected a higher malaria prevalence and reported a malaria prevalence of 31% by nested PCR, compared to malaria prevalence of 19% by microscopy. In view of this, nested PCR is considered to be the most sensitive and specific method of diagnosis, which has been used as a standard reference for comparison in determining the efficacy of RDT and microscopy in malaria-endemic settings. However, it requires highly skilled personnel and a highly equipped state-of-art laboratory with high-cost implications that cannot be met in resource-poor settings. In view of this, microscopy and malaria RDT, which are relatively cheaper than nested PCR, remain very feasible in malaria-endemic, resource-poor settings. 

Currently, malaria RDT is the major method of malaria diagnosis in Africa, where it represented approximately 75% of all diagnostic testing for suspected malaria infection in public facilities in 2017 [39]. Though malaria RDT and microscopy are relatively cheaper than nested PCR, constant monitoring is highly essential to evaluate and validate the efficacy of malaria RDT and microscopy at all times for proper malaria diagnosis, control, and elimination. Unequivocally, our findings apparently revealed that the sensitivity (96.30% vs. 95.06%) and specificity (100% vs. 94.90%) of PfHRP2 RDT and light microscopy in malaria detection were high. This is consistent with the World Health Organization recommendation of 95% sensitivity for PfHRP2 RDT and microscopy at 100 parasites/µL of blood for *P. falciparum* [40]. Similarly, the specificity of 100% recorded for light microscopy was higher than the WHO recommendation of 97% specificity. However, the specificity of PfHRP2 RDT was lower than the 97% specificity recommended by WHO. Apparently, prior studies have revealed the high sensitivity and specificity of PfHRP2 RDT, particularly in high malaria-endemic regions. This is exemplified by the meta-analysis review conducted by [8] on PfHRP2 RDTs, which showed that the sensitivity and specificity were 95.0% and 95.2%, respectively. Similarly, the high sensitivity of microscopy recorded in this study was consistent with the study conducted by Mekonnen [41], who reported the sensitivity of microscopy in the detection of *P. falciparum* and *P. vivax* to be 95.3% and 96.4%, respectively. Similarly, other studies have demonstrated high sensitivity of PfHRP2 RDT and light microscopy in Nigeria [12,29,42] and other malaria-endemic regions [28,43,44]. 

On the other hand, the sensitivity obtained in this study was higher than that obtained in a systematic review and meta-analysis conducted by Yimam [45], who reported sensitivities of 42% and 61% for conventional malaria RDT and ultrasensitive malaria RDT, respectively. The discrepancy in sensitivity observed in this study, compared to others, could be the result of the variation in malaria endemicity across the studies, study population, sample size, and reference tools. Although PfHRP2 RDT has high sensitivity and specificity, the sensitivity and specificity of microscopy were higher. In line with a plethora of reports from previous studies conducted in malaria-endemic regions, light microscopy has been shown to be able to detect approximately 50–200 parasite/μL of blood, which is higher than the detection limit of PfHRP2 RDT, which was observed to be ≥100 parasite/μL of blood [27,46]. 

Apparently, the health officers understood how to use PfHRP2 RDTs correctly without any difficulty, which therefore confirms that PfHRP2 RDTs are simple tests to perform and interpret [47]. The PfHRP2 RDT sensitivity of 95.06% and specificity of 94.90% in our current study were in conjunction with the recent review by Ochola [48], who concluded that the accuracies of the PfHRP2-based RDT test in malaria-endemic regions are usually high, with an average sensitivity of 93%. The sensitivity and specificity documented in our study were higher than the sensitivity of 89% and specificity of 88% reported by Ishengoma [49] in Tanzania. Additionally, our finding is higher than the sensitivity and specificity values of 51.7% and 94.1% reported by Ranadive [50] in Swaziland. In the same vein, a similarly lower sensitivity of 88% and specificity of 76% were reported by Fagbamigbe in Nigeria [42]. Additionally, Landier [51], Hofmann [52], and Girma [53] reported a lower sensitivity of 36.6% in Myanmar, 51% in Papua New Guinea, and 33.9% in Ethiopia, respectively. The high sensitivity recorded in this study could be due to the fact that our study examined symptomatic patients with fevers of ≥37.5 °C, which is likely associated with high parasite density and easily detectable by RDT [49]. It has been confirmed by several studies that PfHRP2 RDT can better detect high-density malaria infection, but with a poor performance in detecting low-density malaria infection, particularly when parasite density is below 100 parasite/μL of blood [54]. Thus, low parasite density has greatly resulted in the low performance of RDT. Therefore, the high sensitivity of PfHRP2 RDT suggested that true malaria cases would rarely be missed, thereby making PfHRP2 RDT a suitable test for malaria diagnosis in high malaria-endemic settings.

While microscopy revealed a zero false-positive result in this study, malaria RDT showed a false-positive rate of ≃5%. Apparently, the false-positive rate of PfHRP2 RDT observed in this study was lower than the false-positive rate of PfHRP2 RDT reported from some other high malaria-endemic regions, such as Zanzibar [55], Ethiopia [56], and Nigeria [12], which have almost the same malaria transmission patterns. Similarly, the false-positive rate was also lower than the false-positive rate of malaria RDT reported from low malaria-endemic settings, such as Swaziland [50]. The false positivity of PfHRP2 RDT observed in this study can be attributed to the presence of PfHRP2 antigen in the blood, long after complete parasite clearance, due to the administration of appropriate doses of the drug of treatment. It has been documented that the PfHRP2 antigenemia can persist for up to 28–42 days, even after successful treatment and parasite clearance from the peripheral blood [57]. Thus, our findings suggest that any patient who has had malaria infection and was treated in a recent time should be given alternative malaria management and avoid the misuse of ACTs. Apparently, false-positive PfHRP2 RDT results could have major public health implications, resulting in overdiagnosis and subsequently overtreatment, thereby jeopardizing antimalarial drugs, which may lead to the development of resistant *Plasmodium* parasites. Additionally, it could result in a significant wastage of drug resources on false-positive individuals. Thus, it will be appropriate to use a better sensitive diagnostic tool for malaria infection confirmation before treatment.

Furthermore, in this study, microscopy showed a false-negative rate of ≃4% among submicroscopic individuals, while PfHRP2 RDT showed a false negativity of ≃5%. Our findings in this study were lower than the false-negative results of microscopy and PfHRP2 RDT obtained in other studies conducted in malaria-endemic settings [12,56]. The false negativity recorded in this study was primarily due to low parasite density, which may have resulted in low antigenemia that was relatively below the detection capacity of microscopist and PfHRP2-based malaria RDT [58]. Obviously, individuals with low parasite density were the most common in this study, accounting for approximately 58.7% of all the individuals examined, and could lead to the presence of PfHRP2 RDT false positivity. Unequivocally, it has been shown that the detection limit of PfHRP2 RDT and microscopy is ≥100 parasite/μL of blood and 50–200 parasite/μL of blood, respectively [27,46,59]. Another possible justification for PfHRP2 RDT false negativity could be the deletion of *pfhrp 2* and *pfhrp 3* genes from the parasite. However, available information revealed that the deletion of *pfhrp 2* and *pfhrp 3* genes from the parasite is very unlikely in this study. [60,61]. Rather, it could be attributed to low parasite density, since five cases with low parasite density of 51–100 parasite/μL were undetected by PfHRP2 RDT, but were detected by microscopy and nested PCR techniques. Additionally, the prozone effect may also lead to false negativity. However, the prozone effect arising from antigen overload, due to the presence of a high parasite density, was unlikely, since the samples examined in our study did not have exceptionally high parasite density. The prozone effect could be defined as a false-negative result, due an overload of either antigens or antibodies arising from high *P. falciparum* parasite density during antigen-antibody immunological reactions [62,63]. Obviously, the public health implication of false negativity is immense because it could lead to the non-treatment or underdiagnosis of malaria-infected individuals, which could eventually cause malaria disease progression to severe malaria and subsequently lead to fatal outcomes if the malaria infection is not detected and treated early enough [2]. Moreover, false negative infection constitutes a significant public health problem, with respect to the malaria control effort, as they eventually serve as reservoirs of malaria disease for the ongoing local transmission of malaria infection [12,63]. Thus, there is need to employ a more sensitive diagnostic techniques, such as microscopy and nested PCR, operated by a well-trained expert, before the administration of drug of treatment [3].

Furthermore, our findings revealed that the sensitivity and specificity of microscopy and PfHRP2 RDT decreased with increases in age. This could likely be due to the low parasite density or submicroscopic malarial infection, which may likely be undetected by microscopy and PfHRP2 RDT techniques [27,46,59]. 

In view of the findings, which revealed that PfHRP2 RDT had a high sensitivity detection level in this study, it could be recommended that PfHRP2 RDT could be employed, along with the gold standard method of microscopy, for better detection and treatment of malaria-infected individuals [25]. Meanwhile, PfHRP2 RDT alone can be used in rural settings, particularly in low- and middle-income countries, where state-of-the-art laboratories, electricity, and experts for microscopy are lacking. This will provide an opportunity for the prompt and accurate detection of malaria parasites and ultimately prevent drug abuse and parasite resistance. While PfHRP2 RDT could be very useful for the fast and easy detection of malaria parasites, some factors are known to reduce efficacy, which needs to be taken into consideration when PfHRP2 RDT is being considered for local detection. These factors encompass the effects of temperature and humidity during transport and storage, quality assurance, parasite density and species, and the recent history of malaria infection and treatment. Similarly, it has been reported that histidine-rich protein 2 deletion could also lead to false-negative results, which may lead to underdiagnosis and subsequently high morbidity and mortality rates [2]. As such, there is a constant need for monitoring the performance of PfHRP2 RDT, in order to ensure accurate diagnosis and prompt treatment, particularly in malaria-endemic regions, such as Nigeria. When all these are put into consideration, the goal of eliminating malaria by the year 2030 may not be far-fetched, ultimately leading to a malaria-free world. 

## 5. Conclusions

Conclusively, although the sensitivity and specificity observed in this study were quite high, false-negative rates of ≃4% and ≃5% were obtained for microscopy and PfHRP2 RDT, respectively. Therefore, besides the routine malaria RDT and microscopy used as the major diagnostic tools in Nigeria, nested PCR and other sensitive diagnostic tools, such as loop-mediated isothermal amplification (LAMP), should be included as additional options where it is feasible. Even though nested PCR tools are relatively costly, with respect to the standard diagnostic tool, their subsequent advantage in malaria control and elimination cannot be underemphasized. Therefore, in order to ensure proper malaria control and elimination in the study area, the use of a more sensitive malaria diagnostic tool is highly imperative.

## Figures and Tables

**Figure 1 pathogens-11-01312-f001:**
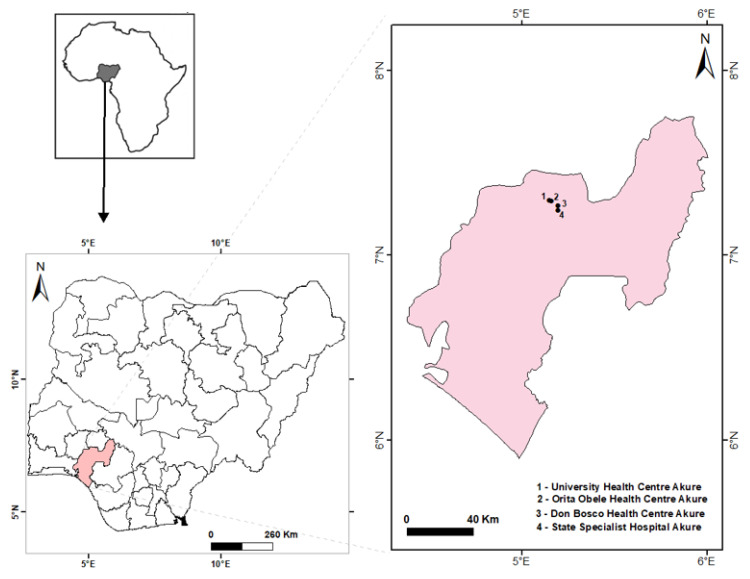
Map of Nigeria showing the sampling sites.

**Figure 2 pathogens-11-01312-f002:**
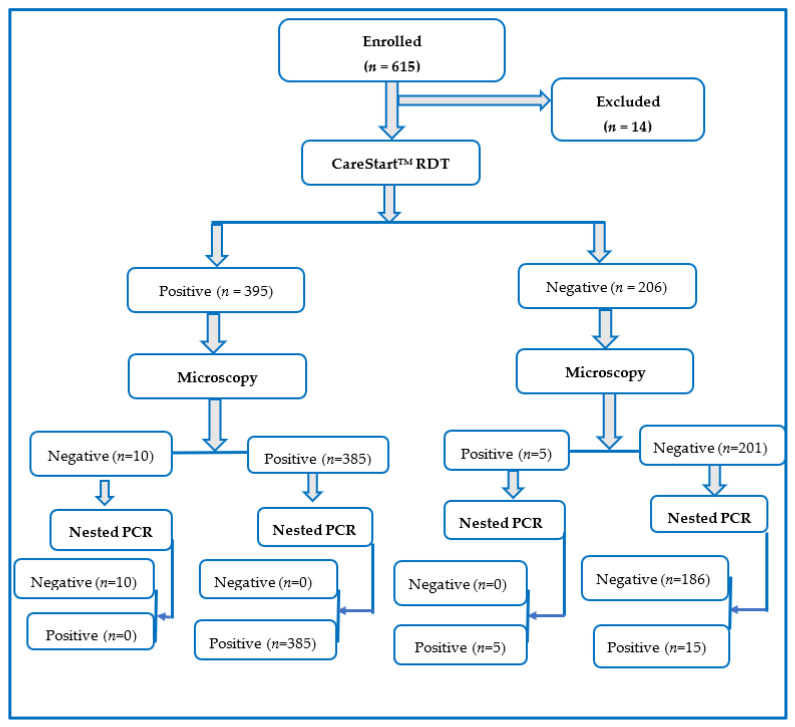
Participants’ flow chart for the diagnosis of *Plasmodium falciparum* infection by microscopy, PfHRP2 RDT, and nested PCR techniques.

**Figure 3 pathogens-11-01312-f003:**
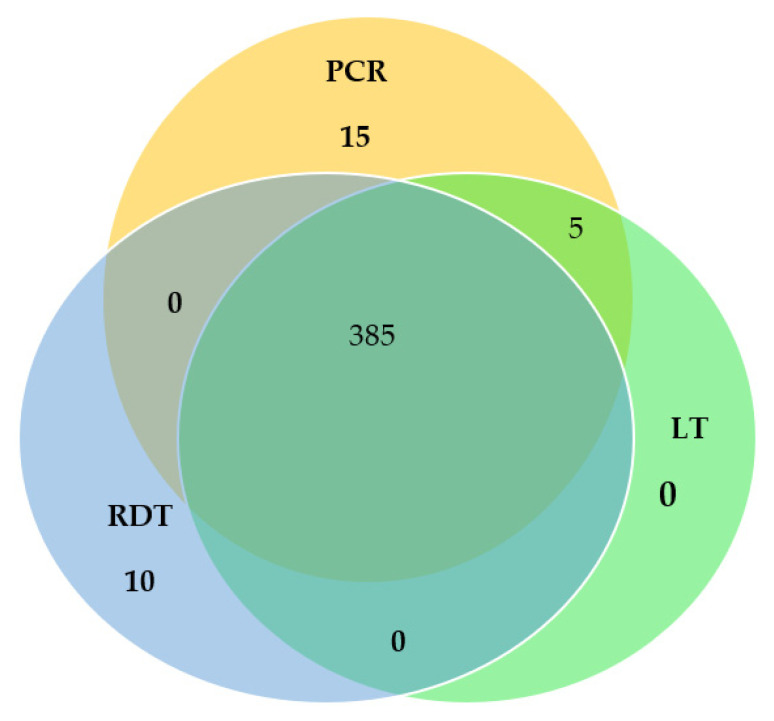
Venn diagram comparing the proportion prevalence of *Plasmodium falciparum* by Light microscopy (LT), Rapid Diagnostic Test (RDT), and nested Polymerase Chain Reaction (nested PCR) techniques.

**Table 1 pathogens-11-01312-t001:** Primers sequences and nested PCR protocol for the detection of *Plasmodium* species.

Genus/Species	PCR Product	Primers	Primer Sequence 5′-3′	Reaction
*Plasmodium* genus	1200 kb	rPLU1	TCAAAGATTAAGCCATGCAAGTGA	Nested 1
		rPLU5	CCTGTTGTTGCCTTAAACTCC	
*Plasmodium* genus	235 bp	rPLU3	TTTTATAAGGATAACTACGGAAAAGCTGT	Nested 2
		rPLU4	TACCCGTCATAGCCATGTTAGGCCAATACC	
*Plasmodium* species	205 bp	FAL1	TTAAACTGGTTTGGGAAAACCAAATATATT	Nested 2
*P. falciparum*		FAL2	ACACAATGAACTCAATCATGACTACCCGTC	
*P. malaria*	144 bp	MAL1	ATAACATAGTTGTACGTTAAGAATAACCGC	Nested 2
		MAL2	AAAATTCCCATGCATAAAAAATTATACAAA	
*P. vivax*	120 bp	VIV1	CGCTTCTAGCTTAATCCACATAACTGATAC	Nested 2
		VIV2	ACTTCCAAGCCGAAGCAAAGAAAGTCCTTA	
*P. ovale*	226 bp	OVA1	ATCTCTTTTGCTATTTTTTAGTATTGGAGA	Nested 2
		OVA2	GGAAAAGGACACATTAATTGTATCCTAGTG	

**Table 2 pathogens-11-01312-t002:** Distribution of participants across the various study locations.

Variables	Study Sites				Total *n* (%)
	Orita Obele Health Center *n* (%)	FUTA Health Centre *n* (%)	State Specialist Hospital *n* (%)	Don Bosco Health Centre *n* (%)	
**Sex**					
Male	-	152 (74.9)	123 (62.4)	66 (65.3)	341 (56.7)
Female	100 (100)	51 (25.1)	74 (37.6)	35 (34.7)	260 (43.3)
**Total**	**100**	**203**	**197**	**101**	**601**
**Age groups (years)**					
≤12	-	1 (0.5)	43 (21.8)	17 (16.8)	61 (10.1)
13–19	17 (17.0)	102 (50.2)	81 (41.1)	30 (29.7)	230 (38.3)
≥20	83 (83.0)	100 (49.3)	73 (37.1)	54 (53.5)	310 (51.6)
**Total**	**100**	**203**	**197**	**101**	**601**

**Table 3 pathogens-11-01312-t003:** Malaria prevalence by microcopy and PfHRP2 RDT, with respect to nested PCR as the reference standard.

		Nested Polymerase Chain Reaction (PCR)		Total
		Positive	Negative	
**Microscopy**	Positive	390	0	390 (64.89)
	Negative	15	196	211 (35.11)
**RDT**	Positive	385	10	395 (65.72)
	Negative	20	186	206 (34.28)
**Total**		**405 (67.38)**	**196 (32.61)**	**601 (100.00)**

**Table 4 pathogens-11-01312-t004:** Diagnostic performance of microscopy and PfHRP2 RDT, with respect to nested PCR as reference.

Test Variables	Microscopy	PfHRP2 RDT
TP (PCR = 405)	390 (100.00)	385 (95.10)
FP (PCR negative)	0	10 (5.10)
TN (PCR = 196)	196 (92.90)	186 (94.9)
FN (PCR positive)	15 (3.70)	20 (4.94)
Sensitivity (95% C.I)	96.30 (93.98–97.74)	95.06 (92.50–96.78)
Specificity (95% C.I)	100.00 (98.08–100.00)	94.90 (90.86–97.21)
PPV (95% C.I)	100.00 (99.02–100.00)	97.47 (95.40–98.62)
NPV (95% C.I)	92.89 (88.60–95.64)	90.29 (85.48–93.63)
Accuracy (%)	97.50	95.01
Cohen’s Kappa (K)	0.94	0.88
Youden Index (J)	0.963	0.8996
*p*-value	<0.0001	<0.0001

**Table 5 pathogens-11-01312-t005:** Sensitivity, Specificity, and Predictive values of microscopy and PfHRP2 RDT by age group, with nested PCR as reference standard.

	Age (Years)	Sensitivity (95% C.I)	Specificity (95% C.I)	PPV (95% C.I)	NPV (95% C.I)	*p* Value
**Microscopy**	≤12	100.00 (92.44–100.00)	100.00 (78.47–100.00)	100.00 (92.44–100.00)	100.00 (78.47–100.00)	<0.0001
	13–19	97.53 (93.82–99.04)	100.00 (94.65–100.00)	100.00 (97.63–100.00)	94.44 (86.57–97.82)	<0.0001
	≥20	94.39 (90.23–96.84)	100.00 (96.74–100.00)	100.00 (97.97–100.00)	91.20 (84.93–95.02)	<0.0001
**Malaria RDT**	≤12	100.00 (92.44–100.00)	100.00 (78.47–100.00)	100.00 (92.44–100.00)	100.00 (78.47–100.00)	<0.0001
	13–19	96.30 (92.16–98.29)	95.59 (87.81–98.80)	98.11 (94.60–99.49)	91.55 (82.76–96.07)	<0.0001
	≥20	92.86 (88.37–95.70)	93.86 (87.87–96.99)	96.30 (92.55–98.19)	88.43 (81.51–92.98)	<0.0001

PPV = Positive Predictive Value; NPV = Negative Predictive Value; C.I = Confidence Interval.

**Table 6 pathogens-11-01312-t006:** Malaria prevalence by microscopy, PfHRP2 RDT, and nested PCR, in relation to parasite density classification.

Parasite Density Levels (Parasite/μL)	Microscopy	PfHRP2 RDT	PCR
<10 (submicroscopic)	0	0	15
51–100 (low)	5	0	5
101–200 (low)	3	3	3
201–500 (low)	124	124	124
501–999 (low)	98	98	98
1000–9999 (moderate)	155	155	155
≥10,000 (severe)	5	5	5
**Total**	**390**	**385**	**405**

## Data Availability

The datasets analyzed during the current study are available from the corresponding author on reasonable request.

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
