# Peer review of "Performance Evaluation of Nested Polymerase Chain Reaction (Nested PCR), Light Microscopy, and Plasmodium falciparum Histidine-Rich Protein 2 Rapid Diagnostic Test (PfHRP2 RDT) in the Detection of Falciparum Malaria in a High-Transmission Setting in Southwestern Nigeria"

_pathogens, 2022, doi:10.3390/pathogens11111312_

Round 1
Reviewer 1 Report
This manuscript evaluated the performance of major malaria diagnostics; i.e. microscopy, Pfhrp2-RDT, and PCR, in the diagnosis of falciparum malaria in a high transmission settings in southwestern Nigeria. The study involved 601 febrile patients and this sample size is advantageous. High detection rate (67.39%) of falciparum malaria was reported. The study reported high sensitivity, specificity and agreement of microscopy and Pfhrp2-RDT methods compared to PCR as the reference method. Also, the study evaluates the performance at different parasite density levels. Although the topic and findings are not novel, the findings are still important to update the performance of major malaria diagnostics at different transmission settings.
Overall, the manuscript is well-written and reported important results. However, some minor corrections and comments need to be addressed before the manuscript can be acceptable for publication. Specific comments are provided below.
SPECIFIC COMMENTS
1. The manuscript needs a more careful proofread to correct some grammatical, punctuation and typographic errors.
2. Title: malaria transmission setting can be indicated. Suggested title: “Performance evaluation of nested ………… in the detection of falciparum malaria in a high transmission setting in Southwestern Nigeria”.
3. Revise the use of abbreviations following first appearance rule. E.g. remove (DBS) in line 22; nested polymerase chain reaction (PCR) in lines 22 & 91 is the second appearance and it can be replaced by “nested PCR”; NPV and PPV abbreviations in abstract can be removed; Plasmodium falciparum in line 68 and line 58; “Pfhrp2 RDT” can be used throughout the text, and etc. Apply throughout the text.
4. Abstract: Results related to performance of the methods at sub-microscopic or different parasite intensity levels can be indicated.
5. Introduction, last paragraph: add few sentences to summarise previous findings on the performance of these methods in Nigeria, and in the study area, if any.
6. Line 27: remove “Similarly,”.
7. Line 30: add “…for falciparum malaria diagnosis ….”.
8. Line 82: “…based on prior reports of plethora of previous studies conducted …”. Please rephrase.
9. Lines 89-90: “Currently, there is need for updated information on the performance of microscopy and malaria histidine rich protein 2 RDT in Nigeria.” This can be removed.
10. Figure 2 can be removed.
11. Line 251: “two-by-two contingency table of”. Can be removed.
12. Table 4: TP of RDT is 95.1.
Author Response
Thank you very much for your insight in the review of this manuscript. The recommended corrections have been made as indicated below.
SPECIFIC COMMENTS
- Reviewer’s comment: The manuscript needs a more careful proofread to correct some grammatical, punctuation and typographic errors.
Author’s response: the manuscript has been proofread to correct the grammatical, punctuation and typographical errors.
- Reviewer’s comment: Title: malaria transmission setting can be indicated. Suggested title: “Performance evaluation of nested ………… in the detection of falciparum malaria in a high transmission setting in Southwestern Nigeria”.
Author’s response: the title has been adjusted as recommended to read as “Performance evaluation of nested polymerase chain reaction (nested PCR), light microscopy and Plasmodium falciparum histidine rich protein 2 rapid diagnostic test (pfhrp-2 RDT) in the detection of falciparum malaria in a high transmission setting in Southwestern Nigeria”.
- Reviewer’s comment: Revise the use of abbreviations following first appearance rule. E.g., remove (DBS) in line 22; nested polymerase chain reaction (PCR) in lines 22 & 91 is the second appearance and it can be replaced by “nested PCR”; NPV and PPV abbreviations in abstract can be removed; Plasmodium falciparum in line 68 and line 58; “Pfhrp2 RDT” can be used throughout the text, and etc. Apply throughout the text.
Author’s response: all the abbreviations have been adjusted as appropriate.
- Reviewer’s comment:Abstract: Results related to performance of the methods at sub-microscopic or different parasite intensity levels can be indicated.
Author’s response: this has been indicated in table 5 with the title “Malaria prevalence by microscopy, RDT and nested PCR in relation to parasite density”. Meanwhile table 6 have been included and discussed to indicate the performance of microscopy and malaria RDT across age group. Thank you sir.
- Reviewer’s comment:Introduction, last paragraph: add few sentences to summarise previous findings on the performance of these methods in Nigeria, and in the study area, if any.
Author’s response: previous findings have been included as suggested. Additional information was added as “For instance, the sensitivity of 87.5% for malaria RDT and 37.5% for microscopy was reported in Lagos and Edo states, Nigeria [14]. Similarly, a sensitivity of 29% and a specificity of 89% were reported in a hospital in Asaba, southern part of Nigeria (15)”
- Reviewer’s comment: Line 27: remove “Similarly,”.
Author’s response: the word “Similarly,” have been removed.
- Reviewer’s comment: Line 30: add “…for falciparum malaria diagnosis ….”.
Author’s response: the phrase “for falciparum malaria diagnosis” have been added.
- Reviewer’s comment: Line 82: “…based on prior reports of plethora of previous studies conducted …”. Please rephrase.
Author’s response: now rephrased as “based on plethora of reports from previous studies”
- Reviewer’s comment:Lines 89-90: “Currently, there is need for updated information on the performance of microscopy and malaria histidine rich protein 2 RDT in Nigeria.” This can be removed.
Author’s response: the statement has been removed as recommended.
- Reviewer’s comment: Figure 2 can be removed.
Author’s response: figure 2 has been removed.
- Reviewer’s comment: Line 251: “two-by-two contingency table of”. Can be removed.
Author’s response: the phrase “two-by-two contingency table of” has been removed.
- Reviewer’s comment: Table 4: TP of RDT is 95.1.
Author’s response: 95.1 has been included in the table as suggested.
Reviewer 2 Report
English revision is required especially in the discussion, check lines 334-335.
It would be interesting to expand the analysis and discussion on the implications of the hyperendemic epidemiological context given the substantial population sample size including different age groups.
Comparing the examined parameters across age groups would be useful in order to evaluate the role of immunity: in particular individuals of the youngest age group may have not yet achieved a sufficient premunition compared to others, thus resulting into different estimates of sensitivity, specificity, PPV, NPV, accuracy and J index due to a higher parasite density.

Author Response
Thank you very much for your insight on the review of this manuscript. The recommended corrections have been made as indicated below.
Reviewer’s comment: English revision is required especially in the discussion, check lines 334-335.
Author’s response: the manuscript has been proofread to correct the grammatical, punctuation and typographical errors.
Reviewer’s comment: It would be interesting to expand the analysis and discussion on the implications of the hyperendemic epidemiological context given the substantial population sample size including different age groups.
Author’s response: Please I will like to state that I have already included and explained the hyperendemic epidemiological context, classification, and implication in a different manuscript that analyse the prevalence, parasite density and risk factors for malarial infection in the study area. Thank you for understanding sir.
Reviewer’s comment: Comparing the examined parameters across age groups would be useful in order to evaluate the role of immunity: in particular individuals of the youngest age group may have not yet achieved a sufficient premunition compared to others, thus resulting into different estimates of sensitivity, specificity, PPV, NPV, accuracy and J index due to a higher parasite density.
Author’s response: Table 6 have been included and discussed to indicate the performance of microscopy and malaria RDT across age group. Thank you sir.